# MULTI-HEAD RAG: SOLVING MULTI-ASPECT PROBLEMS WITH LLMS

## ABSTRACT

Retrieval Augmented Generation (RAG) enhances the abilities of Large Language Models (LLMs) by enabling the retrieval of documents into the LLM context to provide more accurate and relevant responses. Existing RAG solutions do not focus on queries that may require fetching multiple documents with substantially different contents. Such queries occur frequently, but are challenging because the embeddings of these documents may be distant in the embedding space, making it hard to retrieve them all. This paper introduces Multi-Head RAG (MRAG), a novel scheme designed to address this gap with a simple yet powerful idea: leveraging activations of Transformer's multi-head attention layer, instead of the decoder layer, as keys for fetching multi-aspect documents. The driving motivation is that different attention heads can learn to capture different data aspects. Harnessing the corresponding activations results in embeddings that represent various facets of data items and queries, improving the retrieval accuracy for complex queries. We provide an evaluation methodology and metrics, multi-aspect datasets that we release online, and real-world use cases to demonstrate MRAG's effectiveness, showing improvements of up to 20% in relevance over standard RAG baselines. MRAG can be seamlessly integrated with existing RAG frameworks and benchmarking tools like RAGAS as well as different classes of data stores.

## 1 INTRODUCTION

Large Language Models (LLMs) transformed many machine learning tasks using in-context learning abilities. They achieved such accuracy by leveraging an increasing number of parameters, which in recent models have grown to hundreds of billions, making LLM training expensive in terms of both time and resources. It also comes with the danger of leaking confidential data into model weights (Yan et al., 2024; Wang et al., 2024a; Patil et al., 2024). Additionally, continuous training through fine-tuning is necessary to keep LLMs up-to-date. Even using the newest data, LLMs display an ongoing problem of hallucinations (Zhang et al., 2023; Xu et al., 2024c; Huang et al., 2023) by providing factually incorrect information. Retrieval Augmented Generation (RAG) was proposed (Lewis et al., 2020; Guu et al., 2020) in order to address these issues as well as others and make LLMs more trustworthy.

The key idea behind RAG is to enhance the generative model's capabilities by integrating a retrieval system that fetches relevant passages from a large corpus of data. In this setting, when a query is received, the retrieval system first identifies and retrieves pertinent information, which is fed into the generative model's context for a more accurate and relevant response. Instead of the model storing information within its weights, RAG effectively leverages external knowledge, reducing hallucinations (by grounding the LLM reply in reliable sources), and ensuring that responses contain up-to-date knowledge (e.g., by accessing the Internet), all without requiring expensive training.

More specifically, there are two main stages in a RAG pipeline: data preparation and query execution. During data preparation, one constructs a vector database (DB) populated with embeddings and their corresponding data items such as documents. During query execution, one constructs an embedding of that query and retrieves data items in the store with similar embeddings.

Intense recent research efforts have been put into RAG (Gao et al., 2024; Zhao et al., 2024; Hu & Lu, 2024; Huang & Huang, 2024; Yu et al., 2024; Mialon et al., 2023; Li et al., 2022). On one hand, different RAG designs have been proposed, for example RAPTOR (Sarthi et al., 2024), Self-RAG (Asai et al., 2023), Chain-of-Note (Yu et al., 2023), and many others (Abdallah & Jatowt, 2024;

Delile et al., 2024; Edge et al., 2024; Manathunga & Illangasekara, 2023; Zeng et al., 2024; Wewer et al., 2021; Xu et al., 2024b). In general, these schemes focus on making the retrieved data more accurate and relevant to the query. There have also been efforts into benchmarking and datasets for RAG evaluation (Chen et al., 2024b; Xiong et al., 2024; Lyu et al., 2024; Es et al., 2023).

Despite all these advances, we observe that no existing RAG scheme or evaluation methodology explicitly targets an important class of problems that come with a high degree of *multi-aspectuality*. These are *problems that require combining several (potentially many) significantly different aspects in a single query*. As a simple illustrative example of such a query, consider the question "What car did Alexander the Great drive?", and assume that the queried model has not been trained on history. When using RAG, to answer this question accurately, one would retrieve two documents, one describing Alexander the Great and one outlining the history of car manufacturing. However, the embeddings of these two documents could be *far away from each other in the embedding space*. At the same time, such queries are common in different industry settings, as indicated by extensive discussions with our industry collaborators. Imagine a chemical processing plant experiencing an equipment accident. One could use an LLM to find the accident cause, which might require the retrieval of multiple, potentially confidential documents to provide the necessary context. These documents could be related to *different* aspects, for example psychological profiles of workers (*"Was the accident due to mismanaging a worker?"*), equipment purchase records (*"Was some equipment part too old?"*), maintenance (*"Was some equipment part rusty?"*), weather (*"Was there a particularly strong thunderstorm at the accident time that could have caused dangerous power spikes in the grid?"*), or even microclimate (*"Was it too humid for an extended period of time in the production hall?"*). As we illustrate in Section 4, such problems pose challenges for existing RAG schemes and have been unaddressed by modern RAG benchmarking pipelines.

In this work, we propose Multi-Head RAG (MRAG): a scheme that addresses the above problem. Common practice in modern RAG designs is the use of embeddings based on *last-layer decoder block activations*. **Our key idea** is to use instead the activations of the *multi-head attention part of the decoder block* as embeddings. The Transformer architecture can be seen as a pipeline with many (e.g., 96 for GPT-3 (Wang et al., 2024c)) blocks, where a single block consists of an attention module and a feed-forward module. Each individual attention module is *multi-headed*: it consists of multiple parts called heads that learn different sets of weight matrices; see Figure 1 for an overview. It is conjectured that these different heads could capture different aspects of the processed data. We use this as a driving design feature that facilitates capturing the potential multi-aspectuality of the data without increasing space requirements compared to standard RAG, and without *any* fine-tuning or other modifications to the harnessed model (**contribution 1**).

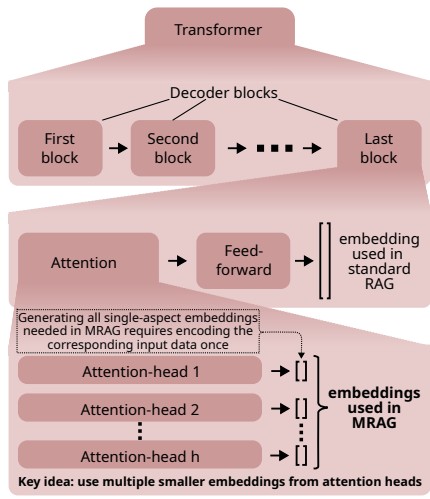

Figure 1: An overview of the decoder architecture, and a comparison of how standard RAG and Multi-Head RAG embeddings are generated.

Such *multi-aspect embeddings* are then directly used for both data items and query representation. Considering multi-aspectuality explicitly comes with challenges. For example, it is unclear how to assess whether a RAG solution does indeed harness multiple aspects when fetching documents. For this, we establish an evaluation methodology as well as a full data construction and query processing pipeline that implements the multi-aspect embedding idea (**contribution 2**). Our datasets facilitate broad evaluation by considering both fully-automatically generated, synthetic data as well as data based on specific industry use cases that show the benefits of MRAG (**contribution 3**). We ensure the relevance of our RAG datasets in real use cases by working directly with tech leaders (e.g., a generative AI division head) from 3 corporations, all of which actively use RAG in their own LLM infrastructures. Our evaluation illustrates the benefits in the relevance of retrieved documents, for example 20% over a modern RAG baseline for fetching multi-aspect Wikipedia articles, and comparable performance for single-aspect queries (**contribution 4**). We also show how MRAG and its benchmarking principles can be seamlessly integrated with both existing RAG solutions and benchmarking frameworks such as RAGAS (**contribution 5**).

## 2 THE MRAG FORMULATION & PIPELINE

We now present in detail the mathematical underpinning of MRAG and its corresponding pipeline.

**Decoder Formulation** We first introduce formally the decoder architecture. We omit, for clarity, unnecessary details such as layer normalizations. The input is a text chunk that consists of $n$ tokens. The output of an attention head $h$ for the $i$th token $x_i$ is defined as (Vaswani et al., 2017) $\text{head}^h(\mathbf{x}_i) = \sum_j w_{ij} \mathbf{v}_j^h$, where $w_{ij} = \text{softmax}\left(\left(\mathbf{q}_i^h\right)^T \mathbf{k}_j^h\right)$, $\mathbf{q}_i^h = \mathbf{W}_q^h \mathbf{x}_i$, $\mathbf{k}_j^h = \mathbf{W}_k^h \mathbf{x}_j$, $\mathbf{v}_j^h = \mathbf{W}_v^h \mathbf{x}_j$. Here, $\mathbf{W}_q^h, \mathbf{W}_k^h, \mathbf{W}_v^h$ are, respectively, learnable query, key, and value projections associated with head $h$, and $\mathbf{x}_j$ is the vector embedding of the $j$th token $x_j$. These outputs get combined to form the output of the $i$th multi-head attention block as $\text{multi-head}(\mathbf{x}_i) = \mathbf{W}_o \text{concat}(\text{head}^1(\mathbf{x}_i), \dots, \text{head}^h(\mathbf{x}_i))^T$, where matrix $\mathbf{W}_o$ is the linear layer that combines the outcomes of all the attention heads. This step is then followed by the Transformer feed-forward layer.

**Standard RAG Formulation** Assume a sequence of $n$ tokens as the input text chunk. The embedding for that chunk is obtained as the activation vector after the *feed-forward* decoder layer for the *last $n$*th token of this chunk, i.e., feed-forward(multi-head($\mathbf{x}_n$)), generated in the *last* decoder block.

**Multi-Head RAG Formulation** The key idea behind MRAG is simple: instead of the *single* activation vector generated by the last *feed-forward* decoder layer for the last token, we harness the $H$ *separate* activation vectors generated by the last attention layer for the last token, *before* processing it via $\mathbf{W_o}$. This can be formulated as a set of embeddings $\mathcal{S} = \{\mathbf{e}_k \forall_k\}$ where $\mathbf{e}_k = \text{head}^k(\mathbf{x}_n)$, which is simply the set of all outputs from the attention heads on the last token $\mathbf{x}_n$ of the input. As processing with multiple heads does not change the size of the output vector, $\mathcal{S}$ has the same space requirements as standard RAG. However, because we capture the separate embeddings before their mixing with $\mathbf{W}_o$, we conjecture that it gives more information about what the *different* parts of the input attend to, facilitating capturing multi-aspectuality.

**Naming** We use the terms "*single-aspect embedding*" and "*multi-aspect embedding*" to refer to, respectively, a small embedding extracted from a single attention head and a collection of all single-aspect embeddings extracted from an attention layer.

### 2.1 OVERVIEW OF THE MULTI-HEAD RAG PIPELINE

We now describe how the above embedding model fits the RAG pipeline. Figure 2 shows a summary of the design. The MRAG pipeline consists of two main parts, dedicated to **data preparation ⓐ** and **query execution ⓑ**. Both parts heavily use the **data store ⓓ** (vector DB).

#### 2.1.1 DATA PREPARATION

When preparing data ⓐ, we populate a data store ⓓ with multi-aspect MRAG text embeddings 〚〛 and their corresponding documents 📄 or text chunks ▦ (MRAG is orthogonal to the type of data being embedded, and while we primarily use chunking of documents in order to reflect modern RAG pipelines, one can also embed whole documents or even other types of data). We create the multi-aspect embedding 〚〛 of each text chunk ▦ using a selected decoder-based embedding model ⓒ (this part is detailed in Section 2.2). The user of the pipeline can plug in their model ⓒ of choice as well as use their input data. We also offer a dedicated synthetic data generator 📚 that can be used to construct multi-aspect input documents 📄 (we detail this part in Section 3) for evaluation purposes.

MRAG stores data differently than standard RAG, where a single embedding 〚〛 points to a single text chunk ▦. For MRAG, each multi-aspect embedding consists of $h$ single-aspect embeddings 〚〛, each pointing to the original text chunk ▦. So the data store ⓓ contains $h$ embedding spaces, each capturing a different aspect of the text. This crucial feature allows MRAG to compare query ⓠ and text chunks ▦ in multiple embedding spaces that capture multiple aspects of the data.

#### 2.1.2 QUERY EXECUTION

During query execution ⓑ, we first generate a multi-aspect embedding 〚〛 of the input query ⓠ, using the selected embedding model ⓒ (details in Section 2.2). Then, we find the nearest multi-aspect embeddings 〚〛 and their corresponding text chunks ▦ in the data store ⓓ using a special multi-aspect retrieval strategy ✿ (detailed in Section 2.3). We ensure that there is **no overhead in latency** due to multiple aspects because computing these different smaller embeddings is done **fully in parallel**. Finally, the retrieved data can optionally be assessed 📊 with novel metrics regarding how well it corresponds to the multi-aspect requirements (detailed in Section 3). As with the data preparation

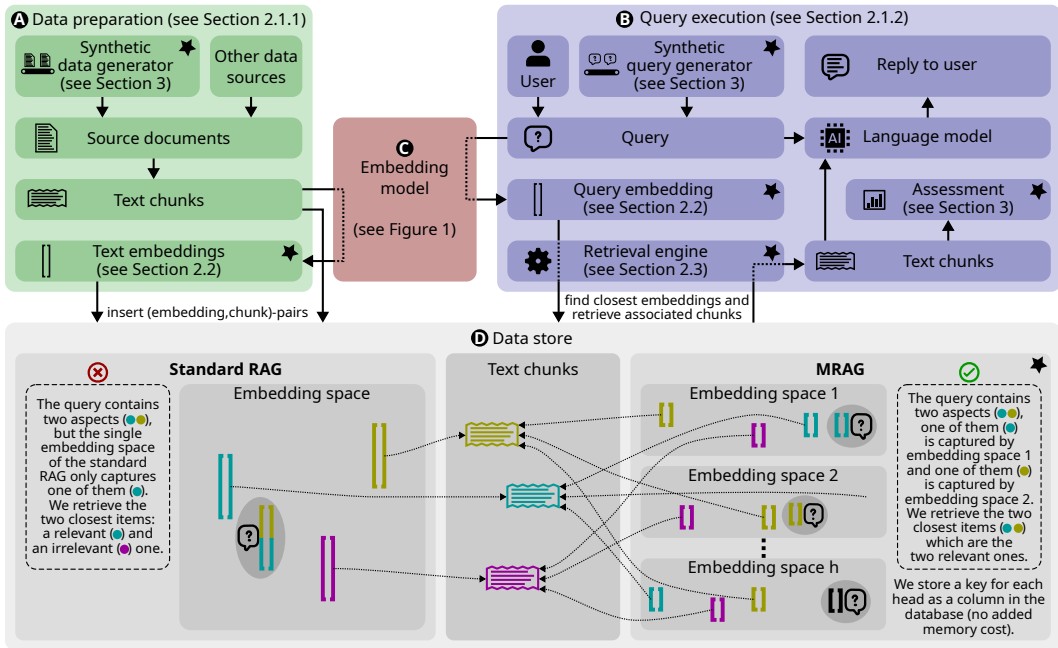

Figure 2: Overview of the MRAG pipeline, consisting of two parts: data preparation Ⓐ and query execution Ⓑ. The embedding model Ⓒ and the data store Ⓓ are used by both parts. The data store Ⓓ contains text embeddings [] linking to text chunks ▥ reflecting three different aspects (cyan, magenta, yellow). Blocks marked by a star ✦ are a novelty of this work.

Ⓐ stage, the query execution Ⓑ stage is flexible, and the user can plug in their models Ⓒ / ▦ of choice and use their own queries ☉. We also offer a dedicated synthetic query generator ⌨ that can be used to construct multi-aspect input queries ☉ (detailed in Section 3) for evaluation purposes.

## 2.2 CONSTRUCTING MULTI-ASPECT EMBEDDINGS []

MRAG can leverage any embedding model with multi-head attention support to construct the multi-aspect embeddings for a given input text. In this work, we consider two embedding models from the MTEB leaderboard (Huggingface, 2024) as potential candidates. Specifically, the SFR-Embedding-Model (Meng et al., 2024) and the e5-mistral-7b-instruct (Wang et al., 2024b), both based on the Mistral 7B architecture with 32 decoder blocks and 32 attention heads per multi-head attention.

While our approach allows for extracting and using the multi-aspect embeddings from *any* decoder block, and from *different* layers *within* a block, we found that multi-aspect embeddings extracted from the last multi-head attention worked best in our experimental setting. We provide further discussion on the carried out experiments in Section 4.

## 2.3 RETRIEVAL STRATEGIES FOR MULTI-ASPECT DATA ⚙

A retrieval strategy determines how we select the closest text chunks from the DB given a multi-aspect embedding of the user query. In general, the MRAG retrieval strategy consists of three steps. First, during data preparation, we **assign importance scores** to all $h$ embedding spaces. Intuitively, these scores capture the fact that different spaces (and the corresponding heads) may be more or less relevant for the used data. Then, during query execution, MRAG starts by applying the traditional RAG retrieval *separately* for each embedding space. This returns a list of $c$ closest text chunks for each embedding space (a total of $h$ lists). Here, we use a special **voting strategy** to pick overall top $k$ out of all $hc$ chunks, using the pre-computed importance scores.

Algorithm 1 details the **construction of importance scores**. It is a heuristic based on extensive empirical evaluation; it gives high-quality results across the tested datasets and tasks. Intuitively, the score $s_i$ of a given head $h_i$ consists of two parts, $a_i$ and $b_i$. $a_i$ is the average of L2 norms of all embeddings in the vector space $i$; it represents how important a given head is: the larger the norms, the more attention was given to this attention head. $b_i$ is the average of cosine distances between all (or a randomly sampled subset, if the user wants to reduce pre-compute time) embeddings in vector space $i$.

Intuitively, $b_i$ is a proxy for measuring the "spread" of vector space $i$: the larger $b_i$, the larger the average angle between different embeddings in this space is. Deriving $s_i$ as a product $a_i \cdot b_i$ ensures that we reward heads with high average attention and high average spread, but simultaneously penalize heads with lower average attention or with low average spread (both $a_i$ and $b_i$ are appropriately scaled).

The used **voting strategy** combines the constructed lists of text chunks from individual embedding spaces into a *single* list of *top $k$* chunks. The strategy is very simple (the corresponding Algorithm 2 is in the Appendix). Each text chunk from a list $i$ of the vector space $i$ has a certain position on this list, we denote this position with $p$. We obtain a weight for this chunk

---

**Algorithm 1** Importance scores for heads.

> **for** each head $h_i$ **do**
>     $a_i \leftarrow 0; b_i \leftarrow 0$
>     $count\_a_i \leftarrow 0; count\_b_i \leftarrow 0$
>     **for** each embedding $e_{ij}$ in $h_i$ **do**
>         $a_i \leftarrow a_i + ||e_{ij}||$
>         $count\_a_i \leftarrow count\_a_i + 1$
>         **for** each embedding $e_{ih}$ **do**
>             $b_i \leftarrow b_i + \text{cosine-distance}(e_{ij}, e_{ih})$
>             $count\_b_i \leftarrow count\_b_i + 1$
>         **end for**
>     **end for**
>     $a_i \leftarrow a_i/count\_a_i; b_i \leftarrow b_i/count\_b_i$
>     $s_i \leftarrow a_i \cdot b_i$
> **end for**

---

as $s_i \cdot 2^{-p}$; $s_i$ is the previously defined importance score of the space $i$. Multiplying $s_i$ with $2^{-p}$ exponentially lowers the significance of less relevant text chunks. Finally, *all* chunks from all lists are sorted using their weights and the top $k$ chunks form the final list.

### 2.3.1 INTEGRATION WITH DATA STORES & OTHER TYPES OF MODELS

MRAG can be seamlessly used with different classes of data stores ◉ and nearest neighbor (NN) search approaches. It can be combined with both the exact and the approximate NN to find the matching (embedding, chunk)-pairs. These two parts of the broader RAG processing pipeline are orthogonal to MRAG. Similarly, MRAG does not depend specifically on the embedding form, as long as it is based on a model that harnesses multi-head attention any such approach that results in a valid embedding can be used. As such, it could also be used with models such as RetroMAE Xiao et al. (2022) and the classic BGE-embeddings Xiao et al. (2022); Chen et al. (2024a).

## 3 MULTI-ASPECT DATASETS, QUERIES, AND METRICS

To assess how well MRAG performs on multi-aspect queries, and to compare it to modern RAG schemes, we need (1) datasets of documents that capture multi-aspectuality, (2) queries to the LLM that touch upon multi-aspectuality and require retrieving documents from the multi-aspect dataset, and (3) metrics that assess how well a RAG scheme retrieves such multi-aspect data. We now describe these three elements. In Section 4, we also briefly discuss real-world data and queries used.

**Multi-Aspect Datasets** We first select conceptually different categories of documents for a synthetic dataset. Here, we harness publicly available Wikipedia articles. In the dataset construction pipeline, the user selects a given number of categories (e.g., countries, board games, historical swords, shipwrecks, etc.) and then, for each category, they sample a specified number of documents. The first part of the document (overview) is used as a text chunk to be embedded. We enforce that each overview must have at least 800 characters, matching commonly used chunk sizes in RAG schemes. We also use multi-aspect **real-world inspired datasets** consisting of NDAs and reports describing industry accidents in chemical processing plants. We ensure the usefulness of these datasets by working directly with tech leaders from 3 corporations that rely on RAG in their in-house LLM-driven report generation and analytics frameworks. Example categories of the legal documents are legal areas (energy law, family law, criminal law, etc.) or document language style (aggressive, mild, neutral, etc.). Examples of accident causes are natural disasters, human mistakes, or lack of proper training. We fully release these datasets to propel RAG research. Details on all three datasets can be found in the Appendix B.2. In our evaluation, we use a total of 13,750 documents.

**Multi-Aspect Query Generation** We also require queries that touch upon a given *number of $n$ aspects*. For example, a query with 10 aspects must contain a question about 10 different documents from 10 different categories. We create such queries by selecting $n$ categories, sampling a document from each selected category (ensuring there are no duplicates overall), and then generating a story that combines these documents, using an LLM (GPT-3.5 Turbo). We construct 25 queries with 1, 5,

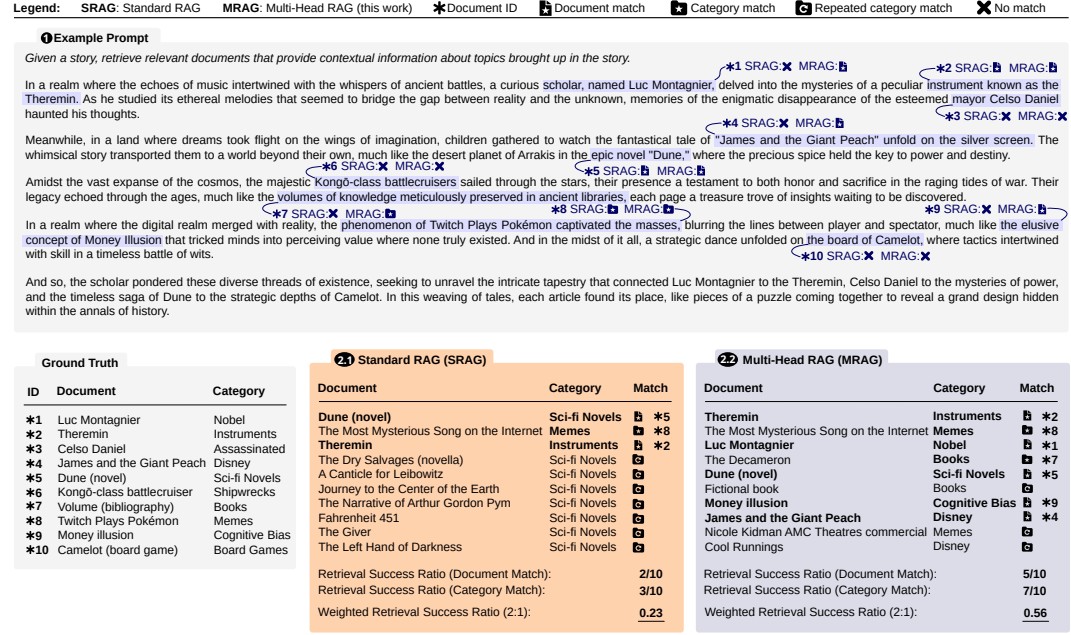

**❶ Example Prompt**

*Given a story, retrieve relevant documents that provide contextual information about topics brought up in the story.*

In a realm where the echoes of music intertwined with the whispers of ancient battles, a curious scholar, named Luc Montagnier, delved into the mysteries of a peculiar instrument known as the Theremin. As he studied its ethereal melodies that seemed to bridge the gap between reality and the unknown, memories of the enigmatic disappearance of the esteemed mayor Celso Daniel haunted his thoughts.   ✴1 SRAG:✖ MRAG:📄   ✴2 SRAG:📄 MRAG:📄   ✴3 SRAG:✖ MRAG:✖

Meanwhile, in a land where dreams took flight on the wings of imagination, children gathered to watch the fantastical tale of "James and the Giant Peach" unfold on the silver screen. The whimsical story transported them to a world beyond their own, much like the desert planet of Arrakis in the epic novel "Dune," where the precious spice held the key to power and destiny.   ✴4 SRAG:✖ MRAG:📄   ✴5 SRAG:📄 MRAG:📄

Amidst the vast expanse of the cosmos, the majestic Kongō-class battlecruisers sailed through the stars, their presence a testament to both honor and sacrifice in the raging tides of war. Their legacy echoed through the ages, much like the volumes of knowledge meticulously preserved in ancient libraries, each page a treasure trove of insights waiting to be discovered.   ✴6 SRAG:✖ MRAG:✖   ✴7 SRAG:✖ MRAG:📄   ✴8 SRAG:📄 MRAG:📄

In a realm where the digital realm merged with reality, the phenomenon of Twitch Plays Pokémon captivated the masses, blurring the lines between player and spectator, much like the elusive concept of Money Illusion that tricked minds into perceiving value where none truly existed. And in the midst of it all, a strategic dance unfolded on the board of Camelot, where tactics intertwined with skill in a timeless battle of wits.   ✴9 SRAG:✖ MRAG:📄   ✴10 SRAG:✖ MRAG:✖

And so, the scholar pondered these diverse threads of existence, seeking to unravel the intricate tapestry that connected Luc Montagnier to the Theremin, Celso Daniel to the mysteries of power, and the timeless saga of Dune to the strategic depths of Camelot. In this weaving of tales, each article found its place, like pieces of a puzzle coming together to reveal a grand design hidden within the annals of history.

**Ground Truth**

| ID | Document | Category |
|---|---|---|
| ✴1 | Luc Montagnier | Nobel |
| ✴2 | Theremin | Instruments |
| ✴3 | Celso Daniel | Assassinated |
| ✴4 | James and the Giant Peach | Disney |
| ✴5 | Dune (novel) | Sci-fi Novels |
| ✴6 | Kongō-class battlecruiser | Shipwrecks |
| ✴7 | Volume (bibliography) | Books |
| ✴8 | Twitch Plays Pokémon | Memes |
| ✴9 | Money illusion | Cognitive Bias |
| ✴10 | Camelot (board game) | Board Games |

**❷.1 Standard RAG (SRAG)**

| Document | Category | Match |
|---|---|---|
| **Dune (novel)** | **Sci-fi Novels** | 📄 ✴5 |
| The Most Mysterious Song on the Internet | **Memes** | 📄 ✴8 |
| **Theremin** | **Instruments** | 📄 ✴2 |
| The Dry Salvages (novella) | Sci-fi Novels | 📄 |
| A Canticle for Leibowitz | Sci-fi Novels | 📄 |
| Journey to the Center of the Earth | Sci-fi Novels | 📄 |
| The Narrative of Arthur Gordon Pym | Sci-fi Novels | 📄 |
| Fahrenheit 451 | Sci-fi Novels | 📄 |
| The Giver | Sci-fi Novels | 📄 |
| The Left Hand of Darkness | Sci-fi Novels | 📄 |

| Retrieval Success Ratio (Document Match): | **2/10** |
|---|---|
| Retrieval Success Ratio (Category Match): | **3/10** |
| Weighted Retrieval Success Ratio (2:1): | **0.23** |

**❷.2 Multi-Head RAG (MRAG)**

| Document | Category | Match |
|---|---|---|
| **Theremin** | **Instruments** | 📄 ✴2 |
| The Most Mysterious Song on the Internet | **Memes** | 📄 ✴8 |
| **Luc Montagnier** | **Nobel** | 📄 ✴1 |
| The Decameron | **Books** | 📄 ✴7 |
| **Dune (novel)** | **Sci-fi Novels** | 📄 ✴5 |
| Fictional book | Books | 📄 |
| **Money illusion** | **Cognitive Bias** | 📄 ✴9 |
| **James and the Giant Peach** | **Disney** | 📄 ✴4 |
| Nicole Kidman AMC Theatres commercial | Memes | 📄 |
| Cool Runnings | Disney | 📄 |

| Retrieval Success Ratio (Document Match): | **5/10** |
|---|---|
| Retrieval Success Ratio (Category Match): | **7/10** |
| Weighted Retrieval Success Ratio (2:1): | **0.56** |

Figure 3: An example query used to evaluate different RAG strategies. We mention the documents to be fetched in the text and then assess the success ratio of different RAG strategies in finding these documents and their categories. We mark exact document matches 📄, category matches 📄, documents that match a category multiple times 📄, and text segments with no matching document ✖. Finally, we show the weighted success ratio for each strategy, taking a 2:1 weighting (prioritizing the exact article matches).

10, 15 and 20 aspects (125 queries in total). An example multi-aspect query sent to the LLM that requires retrieving 10 documents from 10 different categories, is pictured in the top part of Figure 3.

**Metrics** We also design novel metrics to assess how well a given RAG scheme supports multi-aspectuality. For a query $Q$, a used retrieval strategy $S$ (detailed in Section 2.3), and $n$ documents from $n$ categories to retrieve, $Q_{rel}$ denotes the *ideal* set of documents that should be retrieved for $Q$. Then, $S(Q, n)$ is the set of the *actually* retrieved documents. We define the *Retrieval Success Ratio* as $\Xi(Q, n) = \frac{|S(Q,n) \cap Q_{rel}|}{|Q_{rel}|}$, i.e., the ratio of successfully retrieved relevant documents. Moreover, there is a case when a RAG scheme does not retrieve the *exact* desired document, but it still retrieves successfully *some other document* from *the same* category. While less desired, it still increases chances for a more accurate LLM answer following the retrieval. For example, when asking the LLM to determine the cause of an industry accident, fetching the documents in the same category as the accident being queried about, improves the chances for the LLM to give a more relevant answer. To consider such cases, we use another measure, the **Category Retrieval Success Ratio** or $\Xi_c$. It has the same form as $\Xi(Q, n)$ above, with one difference: $S(Q, n)$ is now the set of all the retrieved documents that belong to categories of the ideal desired documents. Finally, to combine these two metrics, we use the **Weighted Retrieval Success Ratio** $\Xi_w$ as $\Xi_w = \frac{w \cdot \Xi + \Xi_c}{w+1}$. By varying $w$, the user can adjust the importance of exact document matches and category matches. An example of using these metrics to assess how well MRAG and Standard RAG capture multi-aspectuality is pictured in the bottom part of Figure 3.

## 4 EVALUATION

We now illustrate the advantages of MRAG over the state of the art.

**Comparison Baselines** We consider three main baselines: **Standard RAG**, **Split RAG**, and **Fusion RAG** (Rackauckas, 2024). The first represents a modern RAG pipeline in which each document uses the activations of the last decoder layer as its embedding. The second is a blend between Standard RAG and MRAG. Specifically, it splits the activation of the last decoder layer in the same way as MRAG and applies a voting strategy. The purpose of Split RAG is to show that *MRAG's benefits come from using the multi-head output as embedding and not merely using multiple embedding spaces*. Additionally, we consider **Fusion RAG** (Rackauckas, 2024), an optional mechanism that

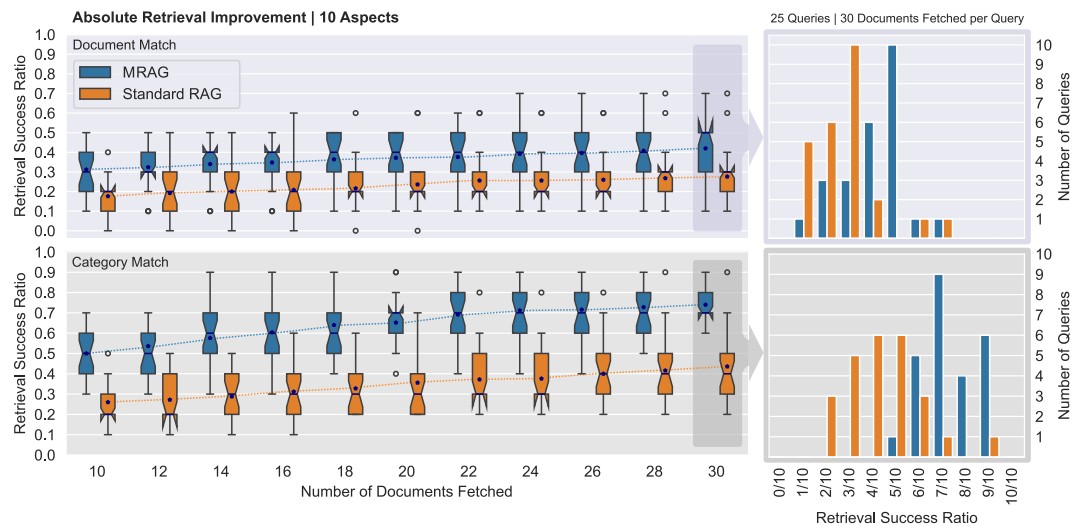

Figure 4: **Retrieval success ratio over 25 queries between MRAG and Standard RAG**; each query uses 10 different aspects. The top part presents **exact document matches**, the bottom part presents **category only matches** (we explain the metrics in Section 3). A histogram is presented for a specific sample to showcase the detailed distribution among the 25 queries (the number of documents fetched for *each* query is 30).

we harness to *further enhance the benefits of MRAG at the cost of additional tokens* (detailed in Section 4.3).

We use **queries** and **metrics** introduced in Section 3. We use the weighted retrieval success ratio with 2:1 weighting, which considers category matches as relevant but prioritizes the exact document matches. Figure 3 shows an example query and metrics usage. Each query requires retrieving a specific number of documents and the corresponding non-overlapping categories which define the ground truth. We fetch the top $k$ documents from a database, where $k$ is the "total number of documents fetched for a tested RAG scheme" (including potentially mismatches). Among these $k$ documents, we search for matches with the ground truth.

**Samples & Summaries** Each data point in our plots corresponds to 25 queries. We present the data using standard boxplots to showcase the distribution. Our primary focus is on the average retrieval performance among those 25 queries.

### 4.1 SUPERIOR PERFORMANCE FOR MULTI-ASPECT QUERIES

We start from the query example in Figure 3 and show first the absolute retrieval performance of MRAG over Standard RAG in Figure 4. We fix the number of aspects present in the queries to 10, and vary the total number of retrieved documents from 10 to 30. MRAG consistently outperforms Standard RAG ($> 10\%$ increase in the retrieval success ratio on average for exact document matches). Moreover, the retrieval performance increase is even more significant on category matches ($> 25\%$ increase in the retrieval success ratio on average). The performance increase is further detailed in the histograms on the right side. Here, for a specific number of documents fetched, MRAG's histogram indicates a better distribution of retrieval success ratios (across all 25 queries).

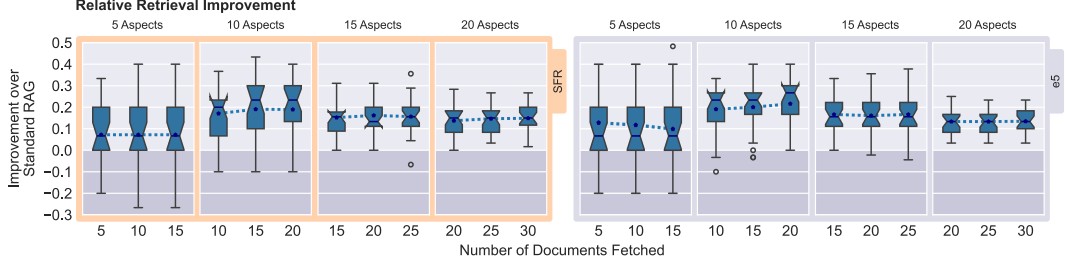

Figure 5: **Relative retrieval improvement of MRAG over Standard RAG** across queries with different numbers of aspects and different embedding models (SFR in the left side, e5 in the right side).

Table 1: **Retrieval success ratio (the exact document match)** for 25 queries **with a single aspect**.

| Documents Fetched | Multi-Aspect Dataset | | | | | Legal Dataset | | | Accidents Dataset | |
|---|---|---|---|---|---|---|---|---|---|---|
| | SFR | | — | e5 | | SFR | | — | SFR | |
| | MRAG | Standard RAG | — | MRAG | Standard RAG | — | MRAG | Standard RAG | — | MRAG | Standard RAG |
| 1 | 24/25 | 25/25 | | 24/25 | 25/25 | | 24/25 | 24/25 | | 25/25 | 25/25 |
| 2 | 25/25 | 25/25 | | 25/25 | 25/25 | | 25/25 | 25/25 | | 25/25 | 25/25 |
| 3 | 25/25 | 25/25 | | 25/25 | 25/25 | | 25/25 | 25/25 | | 25/25 | 25/25 |

Next, Figure 5 shows the relative weighted performance improvement of MRAG with respect to Standard RAG as we vary the number of aspects present in the queries. We show data for two different embedding models (SFR and e5). MRAG consistently outperforms the Standard RAG by 10-20% on average, not only across the number of documents fetched, but also across the number of aspects present in the replies, for both models.

### 4.2 COMPARABLE PERFORMANCE FOR SINGLE-ASPECT QUERIES

We additionally show in Table 1 that MRAG performs on-par with Standard RAG on queries from our multi-aspect dataset where only a single aspect is expected. Hence, our approach does not suffer from significant decrease in performance for single-aspect tasks.

### 4.3 FURTHER IMPROVEMENTS WITH ADDITIONAL TOKENS

We now show that MRAG can be seamlessly integrated with other RAG approaches: We combine MRAG with *Fusion RAG*, representing RAG schemes that use an LLM (additional token cost) for more accurate retrieval. Fusion RAG uses an LLM to create a fixed number of questions about the RAG query. Each question is separately applied through an embedding model using Standard RAG. We apply MRAG's approach to each of these questions and denote the combined scheme as *Fusion MRAG*. Red plots of Figure 6 show that both Fusion RAG and Fusion MRAG perform better than Standard RAG, on average gaining 10 to 30% in accuracy. Fusion MRAG performs consistently better than pure Fusion RAG, indicating that these optimizations can be combined together. However, both Fusion strategies introduce a greater variance than MRAG and additional costs in terms of compute, latency, and tokens.

### 4.4 BENEFITS FROM MULTI-HEAD ATTENTION SOLELY

We also compare MRAG to the Split RAG baseline in Figure 6. The blue plots show the relative weighted performance of MRAG and Split RAG over Standard RAG. MRAG performs better than Split RAG, illustrating that its *high accuracy is due to the actual multi-head part*, and not merely just partitioning the vector and using multiple embedding spaces.

### 4.5 REAL-WORLD WORKLOADS

To further illustrate advantages of MRAG, we also consider two real-word use cases from in-house industry data analytics projects, namely, the synthesis of legal documents and the analysis of causes of chemical plant accidents. The results are in Figure 7. In the former (the left side), the task is to create a document based on user requirements that may be related to different *aspects*, for example to the law being considered (e.g., the British or the US one), the subject (e.g., energetic or civil), the style of the document (e.g., aggressive or mild), etc.. This task is executed with RAG that can fetch documents from a database. In the latter (the right side), the task is to discover a cause of an accident. Here, one also wants to retrieve documents from a database that should be used in the LLM context to facilitate discovering the cause of the accident. The causes are grouped in categories such as utility impact due to severe weather, lack of preparedness and planning, incorrect installation of equipment, lack of maintenance, etc.. Similarly to the previous analyses, we measure the retrieval success ratio over corresponding databases. MRAG offers advantages over other schemes.

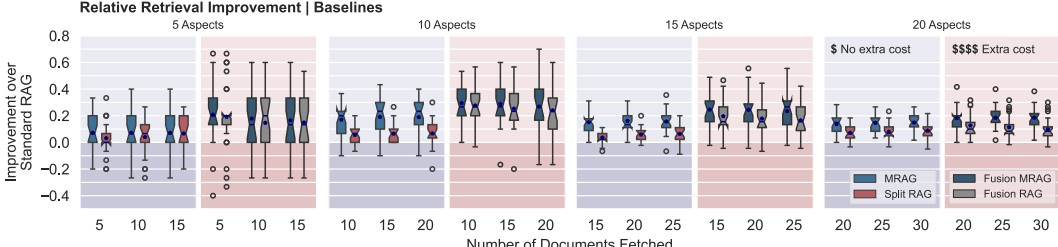

Figure 6: **Relative retrieval improvements of MRAG over Standard RAG** for the SFR embedding model compared with **Split RAG** (the blue plots), and the **relative retrieval improvements of Fusion MRAG over both Fusion RAG and MRAG** (the red plots).

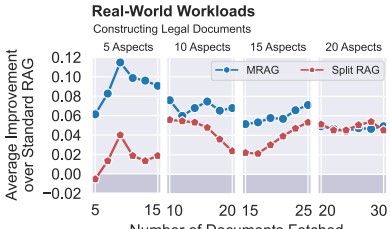
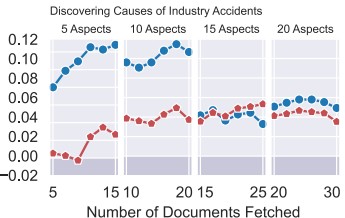

Figure 7: **Average improvement of the retrieval success ratio of MRAG and Split RAG over Standard RAG** for two real-world workloads *constructing legal documents* (left) and *discovering causes of industry accidents* (right).

## 4.6 ADDITIONAL ANALYSES

We also analyze the impact of using embeddings from **different decoder blocks** for MRAG (instead of the last one). Here, we consider taking multi-aspect embeddings from three different layers of the embedding model: after the first multi-head attention block, after multi-head attention block 16 (in the middle of the decoder architecture), and the final multi-head attention. We discover that the last multi-head attention performs the best when compared with the Standard RAG.

We also illustrate selected representative data from a long investigation into two **additional voting strategies** for MRAG. We compare **MRAG (1)** where only the exponential lowering of significance of selected chunks is applied ($w_{i,p} = 2^{-p}$), and **MRAG (2)** which assigns the weight for each text chunk based on the distance between the particular text chunk ($d_{i,p}$) and the query ($q$) ($w_i = \frac{1}{distance(d_{i,p},q)}$). Figure 8 shows that these voting strategies perform worse on average than our selected strategy for MRAG, justifying its design and selection (described in Section 2.3). We also consider two voting strategies for Split RAG, to further deepen the empirical evaluation. **Split (1)** only uses the exponential lowering of significance ($w_{i,p} = 2^{-p}$) and **Split (2)** which uses the same strategy as MRAG ($w_{i,p} = s_i \cdot 2^{-p}$). Figure 8 (on the right) shows that these voting strategies are on-par with each other while being worse than MRAG, further showcasing the advantages of MRAG.

The complexity of the importance score calculation (Algorithm 1) is $\mathcal{O}(n^2)$ where $n$ is the number of the embedded documents; it is dominated by calculating the pair-wise cosine similarity and the calculation of the norm. Please note that this step needs to be done only once for each dataset and is not a bottleneck.

Finally, in addition to the performance evaluation, we also investigated the attention heads of the SFR-Embedding-Mistral model as well as Llama2-7B model (model not fine-tuned for text-embedding tasks). This analysis is presented in Appendix C.

## 5 RELATED WORK

Our work touches on many areas which we now briefly discuss.

Many **RAG schemes** appeared recently (Gao et al., 2024), using the output of the last decoder layer for embedding generation. In contrast, MRAG leverages different embedding spaces of attention heads to focus on different aspects of documents and queries. As such, it can be combined with other schemes to further improve RAG pipelines.

Retrieval is sometimes enhanced by a **cross-encoder reranking** phase (Rosa et al., 2022; Nogueira & Cho, 2020; Nogueira et al., 2020; Li et al., 2021; Gao et al., 2021; MacAvaney et al., 2019). In such solutions, typically after retrieving a set of relevant chunks, they are re-ranked using specialized

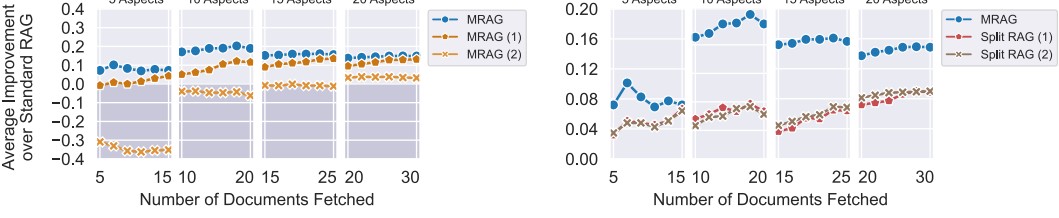

Figure 8: Evaluation of different voting strategies for MRAG and Split RAG.

models. In this work, we focus solely on the first retrieval phase, so MRAG can be seamlessly used in conjunction with such cross-encoders.

**Structure-enhanced RAG** schemes employ different strategies for structuring text to improve retrieval quality. A common idea is to construct a Knowledge Graph from text, which enables retrieval amongst entities and relationships (Jiang et al., 2024; Delile et al., 2024; Hussien et al., 2024; Bui et al., 2024; Xu et al., 2024a). RAPTOR (Sarthi et al., 2024) generates multi-level summaries for clusters of related chunks, building a tree of summaries with increasing levels of abstraction to better capture the meaning of the text. Graph RAG (Edge et al., 2024) creates a Knowledge Graph, and summarizes communities in the graph, which provide data at the different levels of abstraction. All these systems try to improve RAG quality by utilizing additional structures that describe entity relationships or the inner organization of text. Usually, they need a sophisticated preprocessing phase to prepare such structures. MRAG achieves the improvement solely based on the embedding model and has no additional storage requirements, and can be combined with any of these schemes.

## 6 CONCLUSION

Retrieval Augmented Generation (RAG) is pivotal for democratizing access to accurate and relevant outputs from large language models (LLMs). Enhancing the precision and relevance of these outputs is a critical goal, especially given the challenges posed by queries requiring the retrieval of multiple documents with significantly different contents. These complex queries are common across various domains, but existing RAG solutions struggle because the embeddings of the necessary documents can be far apart in the embedding space, complicating their retrieval.

To address this gap, we introduced Multi-Head RAG (MRAG), a novel scheme that leverages the activations from the multi-head attention layer of decoder models instead of the traditional feed-forward layer. This approach is grounded in the insight that different attention heads can capture distinct aspects of the data. By using these diverse activations, MRAG creates embeddings that better represent the multifaceted nature of data items and queries, thus enhancing the retrieval accuracy for complex, multi-aspect queries. The simplicity and versatility of this idea allow it to be seamlessly integrated into any modern RAG pipeline or data analytics framework.

Our comprehensive evaluation methodology, including specific metrics, synthetic datasets, and real-world use cases, demonstrates MRAG's effectiveness. The results indicate a significant improvement in the relevance of retrieved documents, with up to 20% better performance compared to modern RAG baselines. This validates MRAG's potential to handle the intricacies of multi-aspect queries effectively.

Moreover, MRAG proves to be both cost-effective and energy-efficient. It does not require additional LLM queries, multiple model instances, increased storage, or multiple inference passes over the embedding model. This efficiency, combined with the enhanced retrieval accuracy, positions MRAG as a valuable advancement in the field of LLMs and RAG systems. By addressing the challenges of multi-aspectuality in queries, MRAG paves the way for more reliable and accurate LLM applications across diverse industries.

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

# APPENDIX

## A  MODEL DESIGN: ADDITIONAL DETAILS

### A.1  RETRIEVAL STRATEGIES FOR MULTI-ASPECT DATA ✿

---

**Algorithm 2** Voting strategy.

---

$l \leftarrow []$
**for** each head $h_i$ and its score $s_i$ **do**
    find best matching $k$ text chunks
    **for** each chunk $d_{i,p}$ with index $p$ in top $k$
    **do**
        $w_{i,p} \leftarrow s_i \cdot 2^{-p}$
        add tuple $(d_{i,p}, w_{i,p})$ to $l$
    **end for**
**end for**
sort $l$ using weights $w_{i,p}$; return top $k$ elems

---

## B  EVALUATION METHODOLOGY: ADDITIONAL DETAILS

### B.1  COMPUTE RESOURCES

Our experiments were executed with compute nodes containing 4x NVIDIA GH200 and a total memory of 800 GB. In general one GPU with at least 40GB of memory should suffice. We used at most 50GB of storage and the OpenAI API as an external resource. The full experiments took at most three hours of GPU time and the cost for the OpenAI API were at most $15. We carried out additional experiments, which amounted to around 20 hours of GPU time and cost of $25 for the OpenAI API. Additional evaluation was executed with a mix of compute resources including NVIDIA A100 and V100 GPUs.

### B.2  DATASET DETAILS

Table 2: Overview of the structure and the number of documents in the respective datasets.

| dataset | #categories | #topics | #documents | total #documents |
|---|---|---|---|---|
| Wikipedia | 25 | 50 documents per category | | **1250** |
| Legal Documents | 25 | 25 per category | 10 per topic | **6250** |
| Accident Reports | 25 | 25 per category | 10 per topic | **6250** |

## B.3 Prompt Template for the Synthetic Dataset Generation

Table 3: Prompt template for query generation.

Please create a story about the attached <number of articles> articles on the topics <list of titles>.

It is very important that each of the attached articles is relevant to the story, in a way that references the content of the article, not just its title. But please also mention each title at least once. Please make sure that all of the attached articles are relevant to your story, and that each article is referenced in at least two sentences! They do not necessarily have to be referenced in the same order, but make sure no article is forgotten.

Important: Output only the story, no additional text. And do not use bullet points, or paragraphs.

Articles:

————

Article <title>:

<body>

<...>

————

Again, make sure that you reference all the following topics in your story: <list of titles>

## C    Attention Head Analysis

We investigated the attention heads of two models in detail: Llama2-7B and SFR-Embedding-Mistral. We selected these two models for a detailed investigation because the former represents models that are not fine-tuned for text embeddings, while the latter is specifically the text embedding model that we used for our experiments. For each model, we looked specifically at the attention scores within each attention head, i.e., how much attention each head pays to each input token during the inference. Knowing the semantics of the input tokens enables then deriving certain conclusions about multi-aspectuality and attention heads.

We plot selected results in Figure 9. Each heatmap shows the dot-product between key- and value-projections inside a given specified attention head, where line $i$ of a heatmap for attention head h indicates the dot-products between the query-projection of token $i$ and the key-projections of all previous tokens $j < i$ (both models use causal attention).

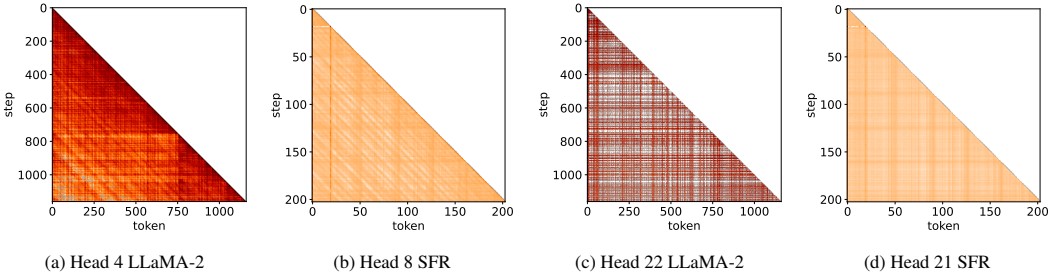

(a) Head 4 LLaMA-2          (b) Head 8 SFR          (c) Head 22 LLaMA-2          (d) Head 21 SFR

Figure 9: Heatmap plots for selected attention heads of the LLaMA-2 7B and SFR-Embedding-Mistral models.

For both models, we found out that the attention patterns vary significantly between the different attention heads. Still, we encountered two distinct patterns. First, the diagonal lines in Figures 9a and 9b indicate that, when processing a certain input token x, elevated attention is paid to some tokens that came a constant numbers of steps before x. We postulate that this pattern is likely beneficial to understanding the overall rhythm of a natural language, allowing the model to better identify which words are semantically connected, and which parts of the input text refer to each other. Second, horizontal and vertical lines in Figures 9c and 9d show that these heads learned to pay attention to specific tokens, regardless of how far apart they are within the input sequence. An intuitive justification for such patterns is the focus on certain semantic aspects of the input sequence.

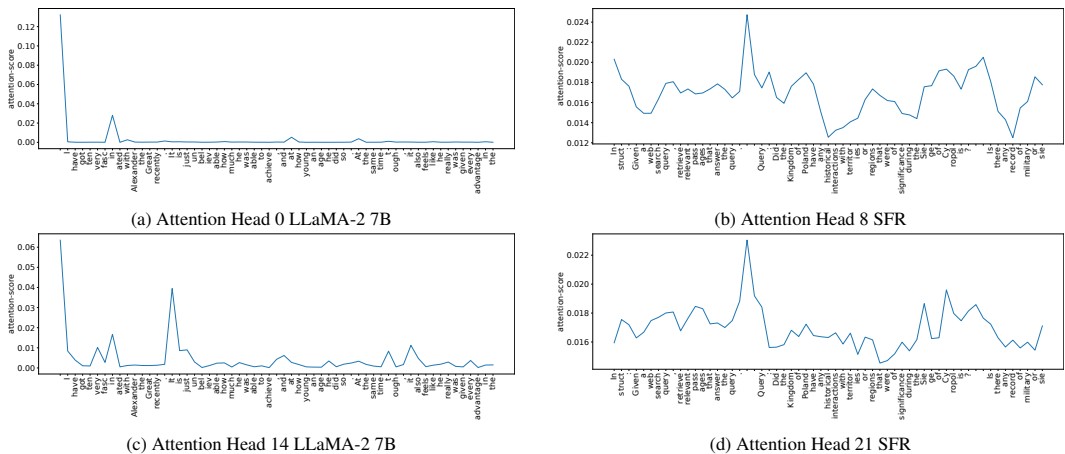

Figure 10: Attention scores for selected attention heads of the LLaMA-2 7B and SFR-Embedding-Mistral models.

We also detail attention scores (after applying softmax) of selected heads in Figures 10 and 11, when the model is processing the last token of its input. We see that some tokens gather a lot of attention from most heads, yet there is always a plethora of passages which are attended differently by any two attention heads. An interesting pattern we encountered was that for the SFR-Embedding-Mistral model (see Figure 11), all heads' attention spiked significantly on the first line-break in the input sequence - either positively or negatively. We conjecture that this is a consequence of how the embedding model was fine-tuned and its intended usage pattern: embedding queries are usually prepended with a retrieval instruction, which is terminated by a line-break. The model likely learnt to summarise the necessary information about this instruction inside the terminating line-break.

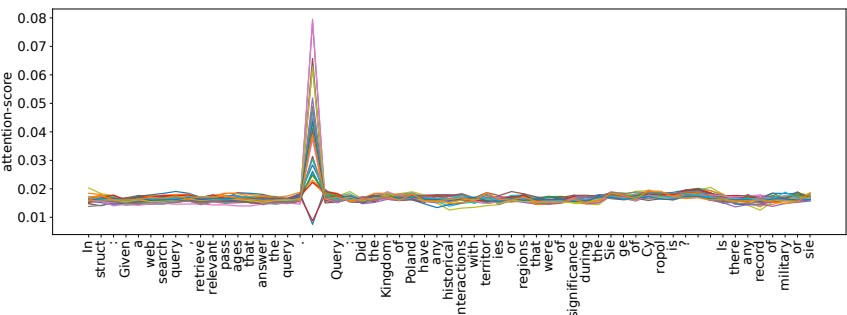

Figure 11: Attention scores for all attention heads of the SFR-Embedding-Mistral model.