# OpenReview forum: "Multi-Head RAG: Solving Multi-Aspect Problems with LLMs"
_ICLR.cc/2025/Conference — Submitted to ICLR 2025_

### Official Review · Reviewer_a1wu · 2024-11-02

**Soundness:** 3
**Presentation:** 3
**Contribution:** 3
**Rating:** 8
**Confidence:** 2

**Summary:**

The authors introduce MRAG, which  is an improved RAG method that enhances LLMs for handling complex, multi-aspect queries. Instead of using a single embedding from the last layer of the decoder, MRAG leverages embeddings from each attention head in the multi-head attention layer.

**Strengths:**

1. The experiments in this paper are thorough and well-executed, covering both synthetic and real-world use cases.  This high level of experimental completeness adds credibility and reliability to the proposed method's performance claims.
2. MRAG introduces a novel approach by leveraging embeddings from each attention head in the multi-head attention layer, a significant departure from standard RAG methods that rely on a single embedding. This innovation enables MRAG to capture diverse dimensions of data within a single query, allowing it to retrieve documents that cover a broader range of relevant topics.

**Weaknesses:**

While the MRAG method improves retrieval accuracy without increasing storage requirements, its multi-head attention multi-aspect embeddings introduce additional computational complexity. In particular, when involving multiple heads, this could result in increased computational costs and potential latency.

**Questions:**

None

---

### Official Review · Reviewer_z3Dt · 2024-11-03

**Soundness:** 2
**Presentation:** 3
**Contribution:** 2
**Rating:** 3
**Confidence:** 2

**Summary:**

The paper addresses the challenge of retrieving multiple documents with different intents for a complex query in RAG systems. The paper introduces MRAG, which leverages the activations of the multi-head attention layer in Transformer models instead of the decoder layer. MRAG shows up to 20% improvement in relevance over standard RAG baselines for multi-aspect queries.

**Strengths:**

The paper identifies a critical problem in RAG and proposes a reasonable solution. It makes a significant contribution by generating a new dataset, query set, and a set of evaluation metrics tailored to assess RAG performance on complex queries with multiple aspects. The paper is well-structured, providing a clear and logical flow of information, although there is still some room for improvement.

**Weaknesses:**

The method proposed in the paper lacks innovation, as the use of multi-head embeddings has been previously explored. The approach of combining results from different embeddings is relatively straightforward and could benefit from the implementation of a more sophisticated heuristic reranker. Additionally, the experiments conducted are insufficient; the use of only 125 queries is inadequate to comprehensively evaluate the performance of the solution for such a complex problem. Furthermore, the advantages over fusion RAG are not clearly articulated.

**Questions:**

How does your approach differ from existing methods that utilize multi-head embeddings other than it is for RAG?

One suggestion is to expand the number of queries in your experiments to provide a more comprehensive evaluation.

---

### Official Review · Reviewer_wEv7 · 2024-11-04

**Soundness:** 2
**Presentation:** 1
**Contribution:** 1
**Rating:** 3
**Confidence:** 5

**Summary:**

This paper addresses the limitation of existing Retrieval Augmented Generation (RAG) solutions in handling queries that require multiple documents with diverse content. It proposes a novel approach called Multi-Head RAG (MRAG), which leverages activations from the multi-head attention layer of Transformers to improve the retrieval of multi-aspect documents. The results indicate that MRAG enhances retrieval accuracy by up to 20% compared to standard RAG baselines while remaining compatible with existing RAG frameworks and datasets.

**Strengths:**

1. The proposed method aims to leverage activations of the Transformer’s multi-head attention layer, instead of the decoder layer, as keys for fetching multi-aspect documents, which is an interesting topic.
2. The authors' harnessing the corresponding activations results in embeddings that represent various facets of data items and queries by multi-head, which seems reasonable.

**Weaknesses:**

1. The paper is not organized clearly, which is not friendly for understanding. For example, there is a lack of details on more detailed comparisons between standard RAG and MRAG.
2. The novelty of the method in the paper seems limited since the multihead idea has been widely studied by previous work [1][2].
[1] Retrieval-Augmented Generation for Knowledge-Intensive NLP Tasks
[2] End-to-End Training of Multi-Document Reader and Retriever for Open-Domain Question Answering

3. The paper lacks the analysis of time complexity as well as space complexity, which is necessary to study the efficiency of the model.
4. The experiments are simple and do not compare with recent methods, such as [1][2], and the experiments utilize GPT-3.5 Turbo, which has been updated and cannot be obtained.
[1] RAGraph: A General Retrieval-Augmented Graph Learning Framework
[2] RQ-RAG: Learning to Refine Queries for Retrieval Augmented Generation

**Questions:**

Please refer to the weaknesses.

---

### Official Review · Reviewer_Sqva · 2024-11-04

**Soundness:** 2
**Presentation:** 2
**Contribution:** 2
**Rating:** 3
**Confidence:** 3

**Summary:**

Existing RAG solutions struggle because the embeddings of the necessary documents can be far apart in the embedding space, complicating their retrieval. To address this gap, MRAG leverages the activation from the multi-head attention layer of decoder models instead of the traditional feed-forward layer. This approach is grounded in the insight that different attention heads can capture distinct aspects of the data.

**Strengths:**

1. This paper establishes an evaluation methodology as well as a full data construction and query processing pipeline that implements the multi-aspect embedding idea.
2.  Evaluation illustrates the benefits in the relevance of retrieved documents, for example 20% over a modern RAG baseline for fetching multi-aspect Wikipedia articles, and comparable performance for single aspect queries.

**Weaknesses:**

1. The motivation of directly using different heads in retriever to retrieve different aspects of the query is not reasonable. The multi-head embedding space is not corresponding to the different aspects in query.
2. The evaluation are conducted on the created dataset and the generalization of dataset is not proved.
3. The compared baselines are limited, other RAG methods should be compared.

**Questions:**

1. Does the retriever in this method need training? If not, how can the different heads in retriever attend to different aspects in your dataset created?
2. The data in Table 1 is confusing. Why almost all datasets and baselines show 100% success retrieval rate. That is contrary to Figure 3.

---

### Official Review · Reviewer_Yg57 · 2024-11-04

**Soundness:** 3
**Presentation:** 3
**Contribution:** 2
**Rating:** 5
**Confidence:** 4

**Summary:**

The authors address an interesting problem: how to handle queries that may require fetching multiple documents with substantially different contents. The authors propose using activations from different heads in the transformer layer as multi-aspect text representations, rather than the final hidden states. Additionally, the authors design a voting strategy to integrate the results from different aspect embeddings.

**Strengths:**

- The research problem is practically significant.
- The idea is simple yet intuitive.
- The author constructed a test dataset specifically for this task and also tested it on real-world use cases.

**Weaknesses:**

- The author’s description of the technical details is unclear. It appears that the activations from different heads in the last layer are directly used as multi-aspect embeddings, but these activations are not trained for retrieval tasks. It hard to believe that they can achieve better results than a trained embedding model.
- The author did not compare their method with other multi-vector representations, such as poly-encoder[1] or ColBERT[2], which can generate multiple embeddings for a text to represent different aspects.
- The evaluation dataset is very small, with only 25 queries per setting, which can lead to significant fluctuations. Minor adjustments could lead to substantial changes in results, making them less convincing.

[1] Poly-encoders: Transformer Architectures and Pre-training Strategies for Fast and Accurate Multi-sentence Scoring
[2] ColBERT: Efficient and Effective Passage Search via Contextualized Late Interaction over BERT

**Questions:**

As mentioned in the Weaknesses, should multi-aspect embeddings be trained, or can they be directly extracted from existing models?

---

### Meta-Review · Area_Chair_Aqh1 · 2024-12-21

**Metareview:**

The authors tackle an intriguing challenge: handling queries that necessitate retrieving multiple documents with significantly diverse content. They propose leveraging activations from various transformer heads as multi-aspect text representations, rather than relying solely on the final hidden states. To combine results from these aspect-specific embeddings, they introduce a voting strategy for integration.

There reviewers have raised several key concerns regarding both the clarity of the presentation and the completeness of the evaluation. Given there is no rebuttal from the authors, I believe this paper is not ready to be published at its current form.

**Additional Comments On Reviewer Discussion:**

No response from the authors

---

### Decision · Program_Chairs · 2025-01-22

Reject